# Reporting of mycetoma cases from skin and soft tissue biopsies over a period of ten years: A single center report and literature review from Pakistan

**Mohammad Zeeshan[1], Saira Fatima[2], Joveria Farooqi[1], Kauser Jabeen[1], Arsalan Ahmed[2], Afreen Haq[1], Muhammad Omer Arif[3], Afia Zafar[1]\***

1 Section of Microbiology, Department of Pathology and Laboratory Medicine, Aga Khan University, Karachi, Pakistan, 2 Section of Histopathology, Department of Pathology and Laboratory Medicine, Aga Khan University, Karachi, Pakistan, 3 Department of Medicine. Dr. Ruth K.M. Pfau Civil hospital Karachi, Pakistan

\* afia.zafar@aku.edu

**Data Availability Statement:** All relevant data are within the manuscript and its Supporting Information files.

## Abstract

### Background

Mycetoma is an important neglected tropical disease associated with debilitation, disfigurement and death if not diagnosed and treated adequately. In Pakistan, mycetoma cases have frequently been diagnosed in histopathology and microbiology laboratories. However, there is scarcity of published data from this country. Therefore, the objectives of this study were to evaluate the frequency and type of mycetoma reported in skin and soft tissue biopsies from a single center over 10 years and review of published literature from Pakistan.

### Method

This descriptive observational retrospective study was conducted at the Aga Khan University Hospital laboratory, Karachi, Pakistan. Laboratory data from 2009–2018 of skin and soft tissue biopsies with positive findings of mycetoma were retrieved from hospital information system. The variables for statistical analysis were age and gender of patient, anatomical site of lesion, residence of patient (geographical location) in the country, etiologic agents of mycetoma and significant gross and microscopic histopathological findings. The data was entered, and descriptive epidemiologic assessment was carried out using MS excel 2013. Geographical information system was used for mapping the location. Literature review of mycetoma cases reported from Pakistan was done on PubMed, Google search and PakMediNet from 1980 till April 2019.

### Result

During ten years of study period, 89 skin and soft tissue biopsies were reported as mycetoma, majority were eumycetoma [n = 66/89 (74%)] followed by actinomycetoma [n = 23/89 (26%)]. Involvement of lower limb was predominantly observed [n = 74/89 (83%)] in which foot had significant contribution [n = 65/74 (88%)]. Only 18 specimens were submitted for

**Funding:** The authors received no specific funding for this work.

**Competing interests:** The authors have declared that no competing interests exist.

microbiological assessment and six grew agents of mycetoma, with *Madurella mycetomatis* reported in only three. Well-formed granuloma formation was observed in only 26%[n = 23/89] of cases. Specific geographical location was not identified, and cases were reported from across the country. From Pakistan, only two original papers and 7 case reports were available in published literature.

## Conclusion

This single center study reports a handful of cases of mycetoma from Pakistan. We conclude that the index of suspicion should remain high among treating surgeons and physicians and clinical laboratories should improve their diagnostic capacity and skills. This will have a great impact on disease outcome and patient's life.

### Author summary

World Health Organization (WHO) has included Mycetoma in the neglected tropical diseases. Epidemiological data of mycetoma cases from Pakistan is not available. Although physicians and surgeons frequently see cases of mycetoma in their clinical practice, but poor socioeconomic conditions of patients restrict laboratory diagnosis such as histopathological and microbiological confirmation of disease. This leads to disability, disfigurement and at times limb loss or mortality. Through this study we are documenting the diagnosed cases of mycetoma from specimens submitted to our laboratory. The diagnosis was mainly made by histopathological examination of samples. It is disappointing to note that mycetoma was not included in the differential diagnosis list of clinicians. Probably, that is the reason that very few samples were submitted to the microbiology laboratory for the culture to find out etiologic agents of eumycetoma and actinomycetoma. Overall, this study has highlighted the gaps in the clinical as well as laboratory diagnosis of this disease and case management.

## Introduction

Mycetoma is a chronic granulomatous infection of skin and subcutaneous tissue that can be caused by fungi (eumycetoma) or bacteria (actinomycetoma). The causative agents are ubiquitous and the disease is endemic in tropical and subtropical regions [1].

The exact prevalence of this disease is largely unknown, though estimates are available from various countries [2]. The clusters of mycetoma cases are reported specifically within certain regions which include Venezuela and Mexico in South America; in Africa it spreads across northwest to north-eastern region, that is, Mauritania, Ethiopia, Senegal, Somalia, and Sudan. In Central Asia, Yemen and Iran are included, and India in South Asia. These regions are collectively labelled as "Mycetoma belt". Its occurrence in different regions varies with environmental temperature and rainfall [3].

Pakistan is an agricultural country with the diverse climate conditions. There are regions that have tropical or subtropical, arid, and deserted climate. Our green and lush north-eastern border consists of agricultural land similar to Indian Punjab. The South-eastern parts, close to Indian desert of Rajasthan, and the western border, adjacent to Iran and Afghanistan, are arid. They have climatic and topographical similarities [4–6]. We speculate that this lack of

information is not a true reflection of burden of disease, probably it is due to poor documentation of diagnosed cases. Given that mycetoma has been included in the list of neglected tropical disease by WHO in 2016, there is a need to highlight the presence of this disease in Pakistan, so, that better patient outcome may be attained by providing more insight to early diagnosis by physicians and laboratories [7].

Therefore, the primary objectives of this study were to evaluate the frequency and type of mycetoma reported in skin and soft tissue biopsies submitted for histopathological examination in our center over 10 years.

The secondary objective was to evaluate the gaps and challenges in the laboratory diagnosis of mycetoma and enlist the important measures needed to overcome the deficiencies in the diagnosis.

We also conducted literature review related to mycetoma cases reported from Pakistan in the last 35 years.

## Methods

Ethics Statement: Approval from ethical review committee of hospital was taken before conducting this study. (ERC:2020-1360-15276)

This descriptive observational retrospective laboratory-based study conducted over ten-year duration from 2009–2018 at the Department of Pathology and Laboratory Medicine, Aga Khan University (AKU), Karachi, Pakistan. The Histopathology section is a very busy unit and receives around 90,000 specimens each year from AKU hospitals, other health care facilities, and through its collection points network located all over the country.

Skin and soft tissue biopsies from all age group and gender with histopathology findings of "mycetoma" were retrieved from integrated laboratory management system (ILMS).

The data was in descriptive form, and therefore, the required variables were entered and assessed using MS excel. The variables included for statistical analysis were age and gender of patient, anatomical site of lesion, geographical location from where the specimen was submitted, type of mycetoma–eumycetoma and actinomycetoma. The histopathological variables included presence of black or white granules on gross examination followed by microscopic findings for the type of infiltrate, abscess and granuloma formation, giant cell reaction, sinus formation and bony involvement. The locations of collection points where specimens were submitted by patient was used in geographic information system (GIS) mapping and considered as reference point of patient's location.

Culture and sensitivity testing were requested and performed only on 18/89 (20%) tissue specimens. As per standard protocol, all biopsy specimens were inoculated in multiple media such as two plates of blood agar, incubated aerobically and anaerobically, chocolate agar in the presence of 5%CO2, two plates of Sabouraud dextrose agar. All plates were incubated for three weeks at $36\pm1$°C except one Sabouraud dextrose agar, which was incubated at 25°C. All organisms were identified phenotypically using mycology bench guide [8,9].

Literature review of mycetoma cases reported from Pakistan was done on PubMed, Google search and PakMediNet using search term that is "mycetoma", "eumycetoma", "actinomycetoma" and "*Madurella mycetomatis*" from 1980 till April 2019. Only cases with discharging sinuses and grains from skin and soft tissue, chronic granulomatous inflammation with evident fungal or bacterial colonies in skin and subcutaneous surgically obtained tissues, dot-in-circle sign in radiology were included. Pulmonary aspergillosis which is also reported in literature as "mycetoma" was excluded.

All case reports, case series, short communication, correspondence, letter to editor and original articles were included and reviewed.

## Results

Total 89 cases of mycetoma were reported in ten-year duration. Majority of the reported cases were eumycetoma 66/89 (74%) while actinomycetoma was seen in 23/89 (26%). The median age of eumycetoma patient was 30.5 years (IQR: 25–42.75). The median age of actinomycetoma patient was 35 years (IQR: 30–45). The male to female ratio in eumycetoma and actinomycetoma was 2.5:1 and 1.5:1 respectively.

Anatomical site involvement is depicted in **Fig 1**. Lower limbs were the most affected anatomical site (83%) while the remaining 17% comprised of upper limb, trunk, head, and neck. For lower limb, contribution of eumycetoma and actinomycetoma was 55/66 (83%) and 19/23 (83%) respectively. For the rest of the body, affecting upper limb, trunk, head, and neck, eumycetoma constituted 7/66 (10%) while cases of actinomycetoma were 4/23 (17%).

Trend of mycetoma cases, reported during 10-years is depicted in **Fig 2A**. The progression was steady, and less than 10 cases of mycetoma were reported every year, however, total number of cases doubled in 2018 with eumycetoma being the predominant type. Culture and sensitivity testing were requested and performed only on 18/89 (20%) tissue specimens.

Out of 18 cultures sent, 6 were positive for agents causing mycetoma. *Madurella mycetomatis* grew in three, *Fusarium* species in two and *Alternaria* species in one biopsy sample. Skin pathogens such as *Staphylococcus aureus* (2/18) and *Streptococcus pyogenes* (1/18) also grew from submitted specimens. Growth of *colonizing skin flora, such as coagulase negative staphylococcus, *Enterobacter* species and *Candida* species were ignored.

The pattern of culture and sensitivity tests requested on suspected mycetoma lesion is depicted in **Fig 2B**, which reflects the inconsistent culture requests from treating clinicians.

Description of histopathological findings reported for both types of mycetoma is elaborated in **Fig 3**. The variables evaluated were the presence of granules on gross examination, type of infiltrate, abscess and granuloma formation, giant cell reaction, sinus formation and bone involvement. The black granules on gross examination were described only in 18% of the specimens.

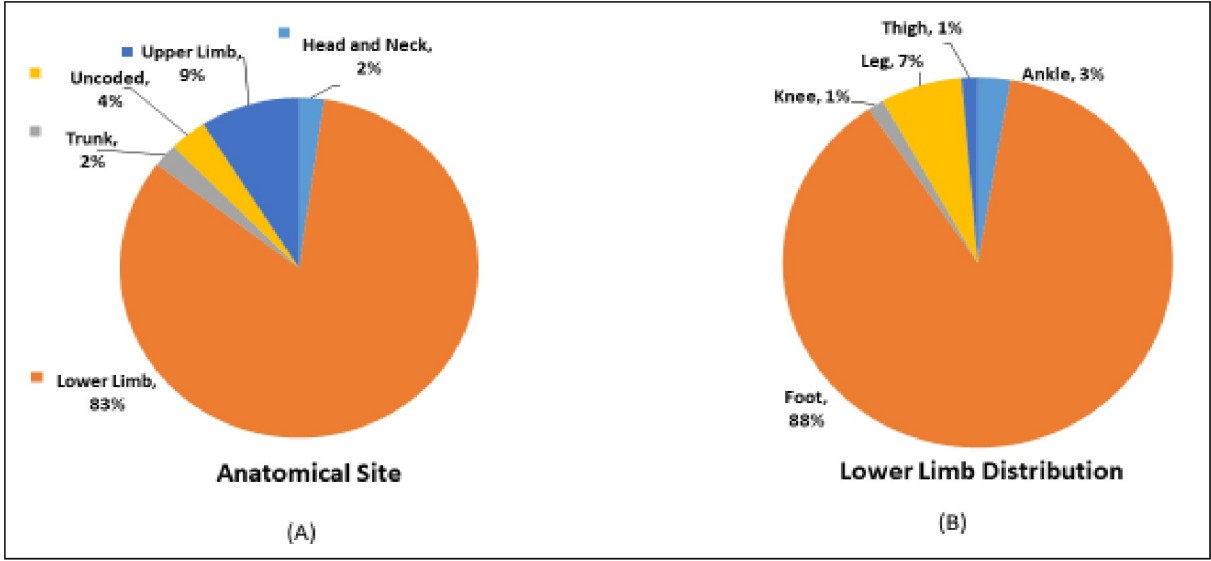

**Fig 1.** Anatomical sites involved (A). Distribution of lesions in lower limb (B).

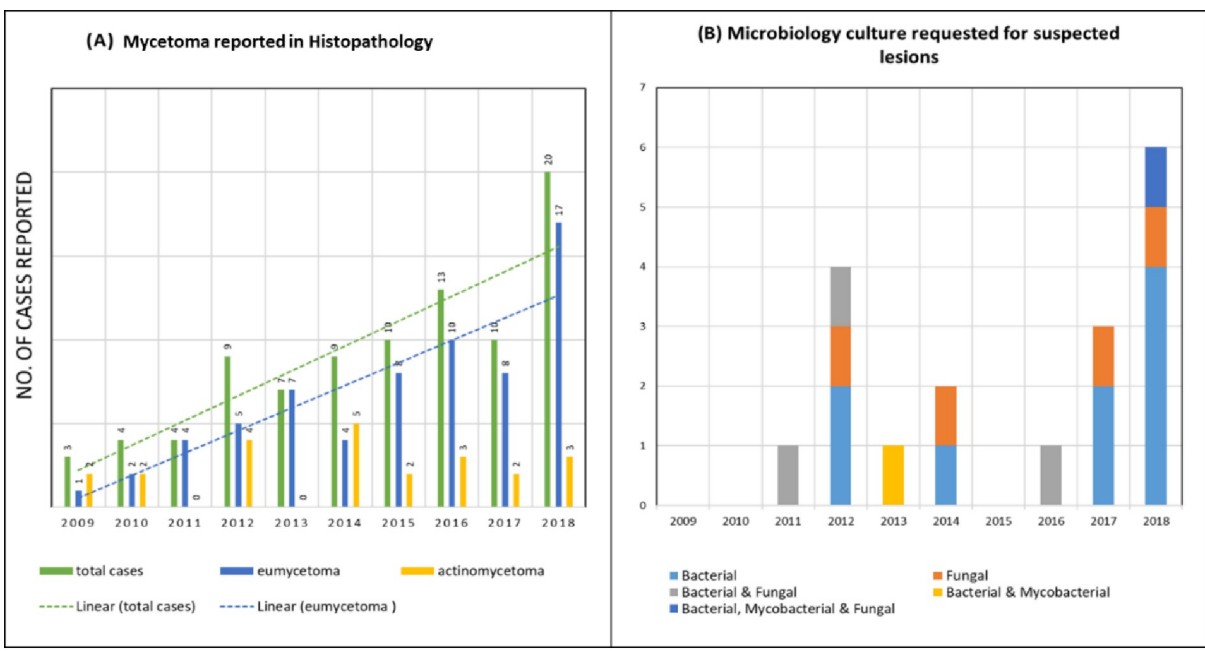

**Fig 2.** (A) Reported cases of mycetoma from section of Histopathology (2009–18). (B) Type of culture requested on submitted specimens (n = 18).

Predominant (80%) tissue reaction was in the form of mixed acute and chronic inflammation. Though well-formed granuloma formation was observed in 26% (n = 23/89) cases, isolated giant cell reaction was present in 33% (n = 29/89) of cases.

Geographical distribution of mycetoma, reported from different cities of Pakistan is highlighted in Fig 4.

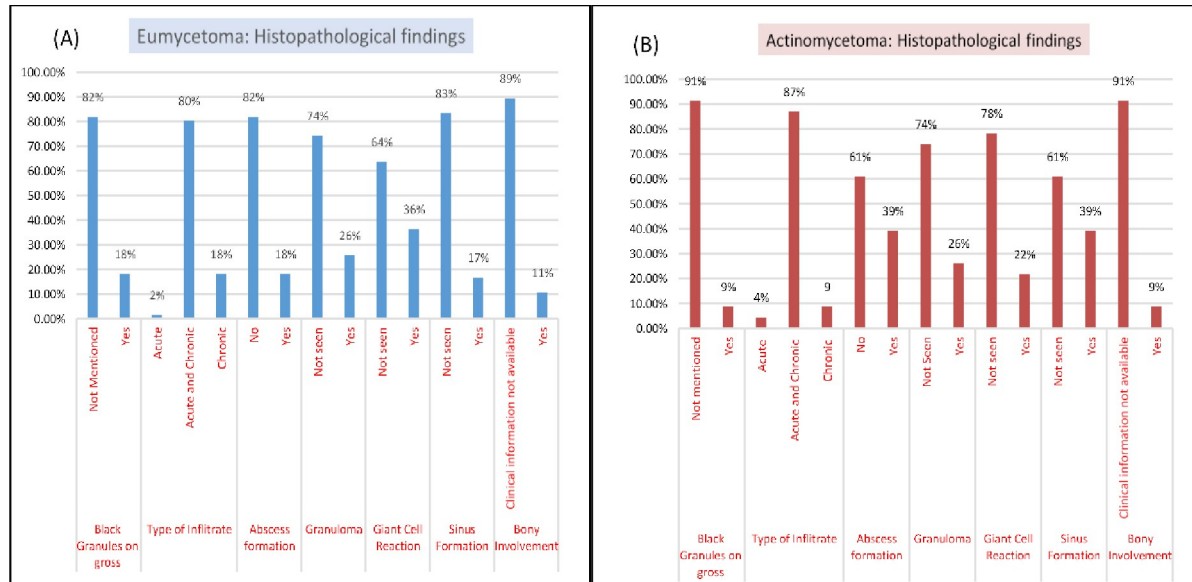

**Fig 3. Histopathology findings in reported cases of mycetoma.**

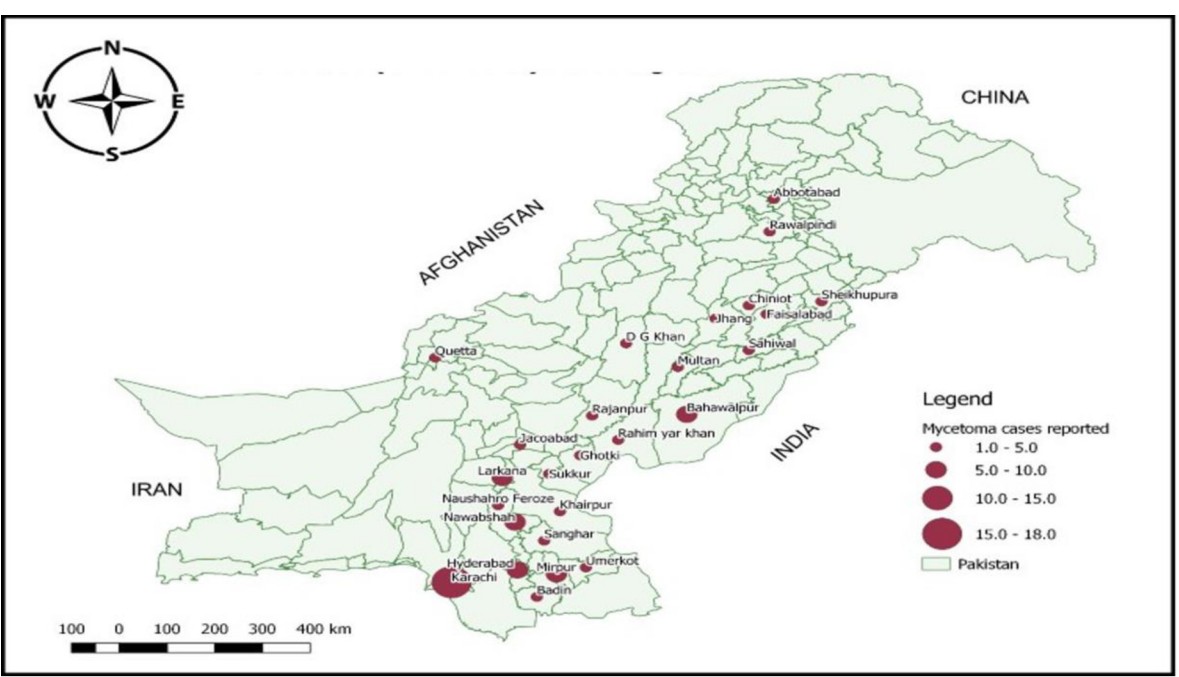

**Fig 4. Geographical distribution of reported cases across Pakistan. (Map is created by author using QGIS software).**

Summary of literature including seven case reports and two original articles is provided in Table 1.

The two original articles with retrospective data analysis were published in local national journals but were not available on PubMed and Google scholar. The study duration in both studies were 4 years. In one of those studies, among 12 cases, histopathologists reported six cases as actinomycetoma, five eumycetoma and one chromomycosis. In the second publication, out of 21 cases, 14 (66%) were reported as eumycetoma and the rest as actinomycetoma. Furthermore, in 7 case-reports, 3 each of eumycetoma and actinomycetoma were reported, while one was reported only as mycetoma without being differentiate into bacterial or fungal type.

## Discussion

In the year 2016, World Health Assembly approved a resolution to include mycetoma in the list of neglected tropical diseases. Information on epidemiological data of disease burden is essential for the developing and implementing public health strategies. Significant data related to mycetoma is available from neighbouring countries such as India and Iran. Both countries share geographical boundaries and have pedological and climatic similarities with Pakistan. Unfortunately, data from Pakistan is scarce and that may be the reason why its name is not included in the list of countries with estimated mycetoma burden [19].

Agricultural activities play an important role in the economy of Pakistan, therefore risk of exposure to microorganisms causing mycetoma are high among farmers and other inhabitants of rural region. Gender parity in agriculture sector is appreciable and young male workers are the main work force. In this study we observed that gender predilection was higher for male in both eumycetoma and actinomycetoma group; this finding is similar to published data from this region and other parts of the world [20].

The most affected anatomical site in both groups of mycetoma was lower extremities followed by upper limb. Majority of the affected are smallholders, at times they live in same living

**Table 1. Literature review of mycetoma reported from Pakistan (1980–2018).**

| Case reports | No | Site of involvement | Geographical location | Mycetoma type | Species | Age in years / Gender | Treatment | Outcome (Ref) | Year of publication |
|---|---|---|---|---|---|---|---|---|---|
| | 1 | Head and neck | Baluchistan | Eumycetoma | *Madurella mycetomatis* | 7/ male | I/V Ampho B for 6 weeks + 6 months of Itraconazole | Extension to cerebrum and death [10] | 2013 |
| | 2 | Foot | NA | Actinomycetoma | *Nocardia species* | 47/ male | I/V Penicillin, PO Co trimoxazole, tetracycline– 3 weeks therapy at 3 different intervals | Cured [11] | 1998 |
| | 3 | Foot | KPK* | Actinomycetoma | NA | 58/ male | NA | Below knee amputation [12] | 2017 |
| | 4 | Arm | KPK* | Actinomycetoma | *Nocardia transvalensis* | 40/ male | PO Co-trimoxazole for 2 weeks | Cured [13] | 1994 |
| | 5 | Face | Punjab | Eumycetoma | *Madurella mycetomatis* | 3/ male | I/V Ampho B for 3 weeks + 6 months of Itraconazole | Cured [14] | 2007 |
| | 6 | Foot | KPK* | Eumycetoma | *Scytalidium dimidiatum* | 20/ male | PO Ketoconazole for 6 weeks | Partially cured -follow up in process [15] | 2003 |
| | 7 | Foot | Punjab | NA | NA | NA / male | NA | Below knee amputation [16] | 2001 |
| **Original article** | No | Study rationale | Study duration and geographical location | # Of cases | Types of Mycetoma | Age range in years / gender | Site | Year of publication | |
| | 1 | Histopathological evaluation of skin and soft tissue swellings after treatment failure with empirical antibacterial, antifungal, and anti-tuberculous therapy | 4 years (1998–2002) KPK* | 12 | 6 Actinomycetoma 5 Eumycetoma 1 Chromomycosis | < 25–65 Male: Female (1.4:1) | Foot [17] | 2004 | |
| | 2 | Retrospective epidemiological data retrieval | 4 years (2012–2016) Sind | 21 | 14 Eumycetoma 07 Actinomycetoma | NA Male: Female 0.75: 1 | Foot [18] | 2014 | |

*KPK = Khyber Pakhtunkhwa

space with their livestock. From crop harvesting to the cattle handling, they work with bare feet and hands. Consequently, these body parts are at higher risk to traumatic inoculation especially feet. These findings are consistent with the available data [21–23], as living and working conditions are similar across the tropical world where this disease is endemic.

As per our data, the majority of the cases were from eastern borders of Sindh and south-eastern borders of Punjab, they have climatic similarities with Indian Rajasthan positioned with the east border of Pakistan. This entire region is arid, barren and have warm desert climate. Due to the severe ecological challenges and under privileged socioeconomic situations their natives rely mainly on animal husbandry to make ends meet. These features provide suitable environment for the propagation of saprophytic soil microorganisms and provide host pathogen interaction usually through penetrating injury with thorns and dry plant. Previously, a single case cohort from Baluchistan province reported similar geo-pedologic conditions as on western border with Iran where mycetoma is prevalent [24].

Traditionally in Pakistan, in the initial phase of the disease, most of the patients explore traditional therapy available in their villages and nearby public sector hospitals. Unfortunately, hospitals based in rural areas are neither properly equipped, nor have expertise to provide

optimum diagnostic results. Therefore, patients after multiple unsuccessful attempts of treatment and with advanced disease are referred to major cities like Karachi in search of better management.

Medical care which includes procedure of surgical biopsy, histopathology and microbiology laboratory services and therapeutic drugs are costly in Pakistan. Most of the time, price of these services is not affordable by many of these deprived patients. That is another reason that patient and their family opt for inappropriate cheaper options in their villages. In the referral centers, the overall cost of management is usually borne by charitable and philanthropist organizations. This problem is evident from our study; as only 20% of the cases had a concomitant request for culture and even then, the yield was very low, only 6/18 (33%). The samples sent from most chronic lesions are poorly collected, frequently contaminated with superficial colonizers, and target inappropriate sampling sites, which do not have adequate grains. The transport conditions are also not ideal: the sample has to travel over long distances, increasing transit time, the tissue samples arrive dry, in unsterile containers, or in contaminated saline. There is also a great need to train surgeons in sending sufficient quantity of sample for culture, which usually arrives as a tissue piece smaller than a centimetre, while most of the tissue is submitted in formalin for histopathology.

The geographical map **Fig 4** is showing that most of the diagnosed cases were registered in Karachi, the largest port city situated in southern coast belt of the country. The environment and infrastructure of this industrial city is not conducive for the acquisition of mycetoma. However, the city of Karachi boasts several large referral centers. Therefore, our data is incorrectly reflecting higher number of mycetoma cases from this town. Unfortunately, due to the retrospective nature of the study we could not investigate the actual place of residence of patients.

The tissue reaction that can be observed in mycetoma cases, vary from Splendor Hoeppli phenomenon, that is adherence of polymorph neutrophils to the granules in background of abscess, mixture of macrophages and giant cells along with neutrophils and formation of well-defined granulomata [25]. In current study, 80% of the cases had mixed acute and chronic inflammation, corresponding to type II tissue reaction. Abscess formation and Splendor Hoeppli phenomenon were noticed in 18% cases. The tissue reaction also indicates the lag period between onset of the disease and time of consult to a physician. We believe that in our study the delay in the tissue diagnosis was due to multiple reasons, such as neglect from patients, limited resources, non-availability of clinical services in remote rural region and ignorance of clinicians that are unable to make a provisional clinical diagnosis and send appropriate specimens for histopathological and microbiological examinations. Non-availability of standardized guidelines for mycetoma cases results in diverse treatment approach, which is evident in the published literature. [26]

Suleiman et. al. described that the lesions of less than 5cm without bone involvement should have wide local excision as primary treatment along with oral antifungals for three months [27]. The lesions between 5–10 cm and bone involvement will benefit with six-month antifungal therapy followed by wide local excision and another six-month antifungal. Patients with lesions greater than 10 cm, secondary bacterial infection and bone involvement may require six-month antifungal therapy with repetitive debridement followed by wide local excision and another six-month antifungal use.

In current study, we are reporting bone involvement in 11% of cases, as per Suleiman et.al recommendations, wider excision of effected tissue and prolong chemotherapy should have been offered to those. We could not gather treatment information for those cases and their outcome due to the retrospective nature of the study.

For the optimum laboratory diagnosis of mycetoma, the process of obtaining specimens in an operating room should be a collective effort between surgeon and the pathologist and microbiology laboratory. However, in resource limited setting this liaison is a huge challenge and specimen submission for culture is often missed by surgical team. This matter is highlighted in our study, and only 18 specimens were found to be submitted for microbiological examination.

Additionally, many physicians don't share the clinical detail of their patient at the time of submission of specimens to microbiology laboratory, which leads to lower yield as agents causing mycetoma grow at a slow rate, need special media, specific incubation temperature and longer incubation than the routine microbiology work. Only growth of *Madurella mycetomatis*, *Fusarium* species and *Alternaria* species were documented. Few cases of smear positive and culture negative cases with no bacterial and fungal growth were also noticed. This reflects fastidious nature of pathogens causing mycetoma.

Additionally, for better yield of pathogens, laboratory capacity building is crucial; technical staff must be trained for processing of specimens, written protocols should be available and must be followed. To exemplify the situation from our study, an eumycetoma causing agent named *Phaeosclera dematiodes* was reported in culture report of a debrided tissue after our study period. Biopsy of same patient was received three years later with suspected diagnosis of chronic infection. We assumed that three years during disease progression, the culture findings of this patient may have remained unnoticed and even the recent attending clinician did not consider fungus in the differential diagnosis.

Literature search revealed that previously from Pakistan, *Madurella mycetomatis* and *Scytalidium dimidiatum* were reported in three cases, whereas *Nocardia* species and *Nocardia transvalensis* in two patients [15]. No uniform treatment regimen and durations for both types of mycetoma were highlighted in these studies. Complete or partial cure were reported in 4 out of 7 cases, in which three patients underwent amputations. In a single case of eumycetoma the disease was extended from face to brain and the patient could not survive. It is disappointing that after extensive literature review, we could not find many case reports with significant microbiological diagnostic support.

One of the limitations in our study was non-availability of gross description on grains morphology. The major reason being that many specimens from various regions of the country and surgical clinics are usually submitted to the laboratory without any clinical history or provisional impression, and in other cases where details are available person performing the gross examination was naïve and could not describe grains morphology. For future, we have planned to improve awareness about importance of gross description of the grains (colour, texture, size) by conducting teaching sessions. We are also planning to develop a gross template for such cases in the section of histopathology and microbiology.

In conclusion, based on to our laboratory data, we are reporting the frequent occurrence of mycetoma cases in Pakistan. This report is highlighting the inadequacy of diagnostic services as well as clinical management. There is a dire need to raise the awareness of the disease in the community, so that farmers and inhabitants of rural region can take protective measures. Additionally, for early recognition of the disease, it is crucial to educate and train health professionals. Better working relations among dermatologists, radiologist, surgeons, microbiologists and histopathologists is also desirable. We consider that this data is representing the tip of an iceberg, therefore it is the right time to initiate mycetoma registry at local and national levels, so that real burden of the disease may be estimated. Only then one can stress upon national health authorities to prioritize this neglected disease by improving patients' access to specialized facilities for better management.

## Author Contributions

**Conceptualization:** Mohammad Zeeshan, Saira Fatima, Joveria Farooqi, Kauser Jabeen, Arsalan Ahmed, Afia Zafar.

**Data curation:** Afreen Haq, Muhammad Omer Arif.

**Formal analysis:** Mohammad Zeeshan, Afreen Haq, Muhammad Omer Arif, Afia Zafar.

**Investigation:** Afia Zafar.

**Methodology:** Afia Zafar.

**Project administration:** Afia Zafar.

**Supervision:** Afia Zafar.

**Writing – original draft:** Mohammad Zeeshan, Afreen Haq, Muhammad Omer Arif, Afia Zafar.

**Writing – review & editing:** Mohammad Zeeshan, Saira Fatima, Joveria Farooqi, Kauser Jabeen, Arsalan Ahmed, Afia Zafar.

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
