## [Decision Letter · Decision Letter 0]

28 Apr 2022

Dear Dr. Zafar,

Thank you very much for submitting your manuscript "Reporting of Mycetoma cases from skin and soft tissue biopsies over a period of ten years: A single center report and literature review from Pakistan." for consideration at PLOS Neglected Tropical Diseases. As with all papers reviewed by the journal, your manuscript was reviewed by members of the editorial board and by several independent reviewers. The reviewers appreciated the attention to an important topic. Based on the reviews, we are likely to accept this manuscript for publication, providing that you modify the manuscript according to the review recommendations. 

Overall, the reviewers are very positive about the work, especially considering it is a retrospective study of a rare entity, as Mycetoma, coming from a country where a few data is available. The reviewers opinions are complimentary and their queries need to be answered point by point in the revised paper. There are considerations about comparing the results with data of Mycetoma in other regions of the planet, and concerns about the lab results. Please, give objective and clear answers about those queries.

Sincerely,

Claudio Guedes Salgado, PhD

Associate Editor

Ahmed Fahal

Deputy Editor

Overall, the reviewers are very positive about the work, especially considering it is a retrospective study of a rare entity, as Mycetoma, coming from a country where a few data is available. The reviewers opinions are complimentary and their queries need to be answered point by point in the revised paper. There are considerations about comparing the results with data of Mycetoma in other regions of the planet, and concerns about the lab results. Please, give objective and clear answers about those queries.

Reviewer's Responses to Questions

**Key Review Criteria Required for Acceptance?**

**Methods**

-Are the objectives of the study clearly articulated with a clear testable hypothesis stated?

-Is the study design appropriate to address the stated objectives?

-Is the population clearly described and appropriate for the hypothesis being tested?

-Is the sample size sufficient to ensure adequate power to address the hypothesis being tested?

-Were correct statistical analysis used to support conclusions?

-Are there concerns about ethical or regulatory requirements being met?

Reviewer #1: objectives are clearly stated, sample size is limited but sufficient as it is a rare disease.

Reviewer #2: Yes is a retrospective, well-conducted study of cases of mycetoma in Pakistan. The inclusion criteria and ethical standards are adequate

**Results**

-Does the analysis presented match the analysis plan?

-Are the results clearly and completely presented?

-Are the figures (Tables, Images) of sufficient quality for clarity?

Reviewer #1: results are clearly presented

Reviewer #2: Yes, there is an analysis of the data and they have been compared with what is reported in the literature of the same country

**Conclusions**

-Are the conclusions supported by the data presented?

-Are the limitations of analysis clearly described?

-Do the authors discuss how these data can be helpful to advance our understanding of the topic under study?

-Is public health relevance addressed?

Reviewer #1: data support the conclusions. limitations as mentioned by authors,it is single center

Reviewer #2: If they provide correct conditions and it is a condition that is important to know about mycetoma in this part of the world, because recently there are no communications

**Editorial and Data Presentation Modifications?**

Reviewer #1: Authors herein conducted an observational retrospective study in a single center to evaluate the frequency and type of mycetoma.Overall article is well written.Would consider comparing the current findings to neighboring countries as well how the findings compare to globally.

Reviewer #2: Minor revison

**Summary and General Comments**

Reviewer #1: Authors herein conducted an observational retrospective study in a single center to evaluate the frequency and type of mycetoma.Overall article is well written.Would consider comparing the current findings to neighboring countries as well how the findings compare to globally.

Reviewer #2: Overall it is a well conducted study of mycetomas (about 90) in Pakistan

Some doubts that they generate in me, because they do not indicate it clearly in the methodology, are the following:

- The number of positive cultures is very low (20%), give an explanation because the majority of agents grow in the usual media

- What was the method of identification of Nocardia species?

- How do they come to identify Scytalidium? what color was the grain? because you have to remember that there is a pigmented variant that is Neoscytalidium

- By morphology of the grains it is possible to suggest types of actinomycetes such as Actinomadura and Streptomyces and not just leave it in general

PLOS authors have the option to publish the peer review history of their article (what does this mean?). If published, this will include your full peer review and any attached files.

Reviewer #1: Yes: K.M Baradhi

Reviewer #2: Yes: Alexandro Bonifaz

Figure Files:

Data Requirements:

Reproducibility:

References

---

## [Editor Report · Decision Letter 1]

24 Jun 2022

Dear Dr. Zafar,

We are pleased to inform you that your manuscript 'Reporting of Mycetoma cases from skin and soft tissue biopsies over a period of ten years: A single center report and literature review from Pakistan.' has been provisionally accepted for publication in PLOS Neglected Tropical Diseases.

Best regards,

Ahmed Fahal, FRCS, FRCSI, FRCSG, MS, MD, FRCP(London)

Deputy Editor

Ahmed Fahal

Deputy Editor

---

## [Editor Report · Acceptance letter]

26 Jul 2022

Dear Dr. Zafar,

We are delighted to inform you that your manuscript, "Reporting of Mycetoma cases from skin and soft tissue biopsies over a period of ten years: A single center report and literature review from Pakistan.," has been formally accepted for publication in PLOS Neglected Tropical Diseases.

Best regards,

Shaden Kamhawi

co-Editor-in-Chief

Paul Brindley

co-Editor-in-Chief
